# Challenges and Opportunities of Force Feedback in Music

Christian Frisson [1,*,†] and Marcelo M. Wanderley [2,*,†]

1. Society for Arts and Technology, Metalab, Montreal, QC H2X 2S6, Canada
2. Input Devices and Music Interaction Laboratory (IDMIL), Centre for Interdisciplinary Research in Music Media and Technology (CIRMMT), McGill University, Montreal, QC H3A 0G4, Canada
* Correspondence: christian@frisson.re (C.F.); marcelo.wanderley@mcgill.ca (M.M.W.)
† These authors contributed equally to this work.

**Abstract:** A growing body of work on musical haptics focuses on vibrotactile feedback, while musical applications of force feedback, though more than four decades old, are sparser. This paper reviews related work combining music and haptics, focusing on force feedback. We then discuss the limitations of these works and elicit the main challenges in current applications of force feedback and music (FF&M), which are as follows: modularity; replicability; affordability; and usability. We call for the following opportunities in future research works on FF&M: embedding audio and haptic software into hardware modules, networking multiple modules with distributed control, and authoring with audio-inspired and audio-coupled tools. We illustrate our review with recent efforts to develop an affordable, open-source and self-contained 1-Degree-of-Freedom (DoF) rotary force-feedback device for musical applications, i.e., the *TorqueTuner*, and to embed audio and haptic processing and authoring in module firmware, with *ForceHost*, and examine their advantages and drawbacks in light of the opportunities presented in the text.

**Keywords:** haptics; force feedback; musical interaction; computer music

## 1. Introduction

Digital musical instruments (DMIs) feature high-resolution gesture sensing and audio output but poor gestural and bodily feedback compared to traditional acoustic instruments, which passively produce and transfer vibrations from strings to fingers; kinesthetic feedback from drums to the fingers, hands, forearms and arms of drummers; and the coupling of air columns between wind instruments and their players. In this article, we examine how DMIs can produce dynamic feedback to the sense of touch of their players, that is, through haptic feedback.

The term *haptics* involves both *touch* and *force feedback*. Touch feedback, specifically *vibrotactile* feedback, has been the focus of much research and development in the last two decades using devices that cause vibrations felt by mechanoreceptors in the skin. A review of the types of mechanoreceptors and their functions is available in Halata and Baumann (2008). This paper focuses on force-feedback applications in audio, music and media control. Touch or vibrotactile feedback is discussed in length in other papers that are part of the same special issue, *Feeling the Future—Haptic Audio*, as our article, as well as in a recent reference edited by Papetti and Saitis (2018). Burdea and Coiffet (2003) define *force feedback* as a simulation that "conveys real-time information on virtual surface compliance, object weight, and inertia. It actively resists the user's contact motion and can stop it (for large feedback forces)".

Research in FF&M intersects with research on *mulsemedia* (or *multisensory multimedia*) systems, as defined by Covaci et al. (2018), and it faces similar challenges to those reviewed by Ghinea et al. (2014). These include the difficulty of implementing systems, as reviewed by Saleme et al. (2019), and the need for authoring tools, as elicited by Mattos et al. (2021).

Force-feedback devices typically use electrical motors to deploy (output) forces based on position inputs. This design—position in and force out—is known as *impedance control*. Devices, therefore, can be characterized in terms of the number of inputs and outputs they possess.

*FF&M*

Research on force feedback applied to music is not recent, with some of its early contributions dating back to the late 1970s. Although a body of work has been developed over the years focusing on measurements, models and applications, musical force feedback has never become widespread. Despite its longevity, it has been impeded by factors such as (rather exorbitant) hardware costs, software limitations (drivers), fast hardware and software obsolescence, and a lack of accessible platforms for prototyping musical applications. Disruptive force-feedback musical application is yet to come.

That being said, the simulation of complex performer–instrument interactions in music is a promising research direction that aims to understand musicians' highly skilled control strategies developed over years of intensive training.

In recent years, several works have addressed several aspects of this topic and will be discussed in the next section.

## 2. Previous Works

In this section, we review previous works related to FF&M. Firstly, hardware devices are reviewed, followed by software environments.

### 2.1. Force-Feedback Devices

The availability of general-purpose force-feedback devices that can be used in real time to provide the closest similarity to an actual instrumental situation is a significant issue, both in terms of the cost of such devices (typically several thousand dollars) and the relatively rapid obsolescence of communication protocols used by them (e.g., communication using the parallel port in some of the older models), which limits the usefulness of such investments.

As for device-specification requirements for FF&M, a large workspace is typically desired (both in translation and in rotation). Strong motors are also required to present rigid walls (especially necessary in the percussive case), and low tip inertia and friction are needed to increase the transparency of the device vis-à-vis the simulated action.

Common devices used in force-feedback research, including musical applications, are typically 3-DoF devices in the form of a stylus or spherical end effector, which provide three output forces in the X, Y and Z axes. Devices such as the *Novint Falcon* measure positions in 3-DoF and offer mapped translation. Others measure positions in 6-DoF—3-DoF in translation and 3-DoF in rotation between the stylus and the arm of the end effector—but still provide only a 3-DoF force output, one example being the *3D Systems' Touch X* (formerly *SensAble's Phantom Desktop*). Devices that output 6-DoF with forces in the X, Y and Z axes, as well as torques around the three axes, are more expensive but are relatively common, with 6-DoF positional sensing available on devices such as *3D Systems' Phantom Premium* or *MPB Technologies' Freedom 6S*.

Apart from the specifics of positional sensing and force feedback, devices differ in two main ways. Firstly, they differ in the usable workspace they provide. The larger the workspace volume, the more expensive the device tends to be, and typically they display lower output forces. Secondly, devices also differ in mechanical construction, which can be 'serial' or 'parallel'. In serial devices, the three output motors are connected to the end effector through a common structure, whilst in parallel devices, each motor connects directly to the end effector. The *Touch X*, *Phantom Premium* and *Freedom 6S* are serial devices, whilst the Falcon is a parallel device.

Further, the *Touch X*, *Phantom Premium* and *Freedom 6S* are three examples of variable workspace and force distributions based on data collected for and categorized in *Haptipedia*

by Seifi et al. (2019); the *Touch X* has a small translational workspace of $16 \times 12 \times 12$ cm$^3$ and a large rotational workspace of $360 \times 360 \times 180$ deg$^3$ and outputs translational forces of 7.9 N (peak) and 1.75 N (constant). The *Phantom Premium* has a large translational workspace of $82 \times 59 \times 42$ cm$^3$ and a large rotational workspace of $330 \times 330 \times 220$ deg$^3$ and outputs translational forces of 22 N (peak) and 3.00 N (constant). Meanwhile, the *Freedom 6S* has a medium translational workspace of $33 \times 22 \times 17$ cm$^3$ and a medium rotational workspace of $340 \times 170 \times 130$ deg$^3$ and outputs translational forces of 2.5 N (peak) and 0.60 N (constant).

Figure 1 shows several commercial devices used in musical applications.

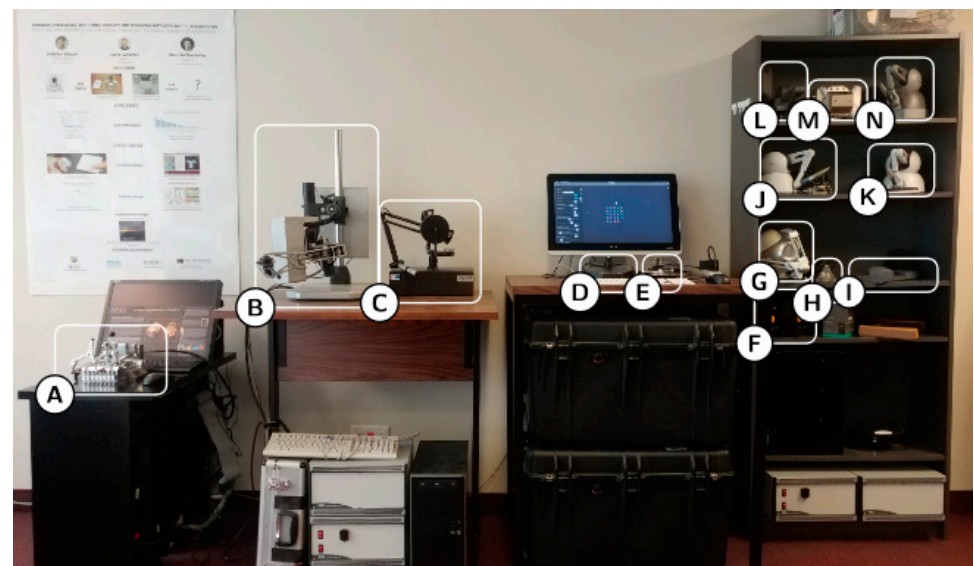

**Figure 1.** Several force-feedback devices used in force-feedback musical applications at the Input Devices and Music Interaction Laboratory, McGill University, Canada (IDMIL). From bottom-left to top-right: (A) *ACROE ERGOS*; (B) *MPB Technologies Freedom 6S*; (C) *SensAble Phantom Premium*, (D) and (E) two *Haply Pantographs*; (F) two *FireFaders* built at *IDMIL*; (H) *Novint Falcon* and (G) removable end effector; (I) *Logitech WingMan* mouse; (J) and (K) two *SensAble Phantom Omni*; (L) *SensAble Phantom Desktop*; (M) a second *ACROE ERGOS*; and (N) *3D Systems Touch*.

One interesting example of a force-feedback device is the *ERGOS*, a high-quality, variable-DoF device developed by the *Association pour la Création et la Recherche sur les Outils d'Expression* (*ACROE*). The *ERGOS* actuator consists of "a stack of flat moving coils that are interleaved with flat magnets", as explained by Florens et al. (2004).

The *ERGOS* is innovative in several aspects as explained by Cadoz et al. (1990) as follows: (a) it consists of multiple 1-DoF *sliced motors* (motors sharing a single magnetic-polarization circuit for use as motor modules in a force-feedback keyboard), which share a common magnetic field, allowing for individual sliced motors of a reduced size; (b) several sliced motors can be combined in a single *ERGOS* device (4, 6, 12 or more motors); (c) individual motors can be connected through mechanical add-ons to create integral 2- to 6-DoF effectors; and (d) it has been primarily designed with artistic applications in mind.

The *ERGOS* was used in several artistic/musical projects at *ACROE*, e.g., "pico..TERA" by Cadoz et al. (2003), as well as by Sinclair et al. (2009) and Tache et al. (2012).

Several force-feedback devices, either generic or specifically designed for musical applications, have been used over the last several decades to simulate instrumental actions. We now review these force-feedback devices, initially with respect to the number of DoF they provide.

### 2.1.1. 1-DoF Devices

Devices of 1-DoF are very useful as they yield detailed explorations of haptic effects in constrained situations. Several applications can be simulated with 1-DoF devices, for instance, feeling bumps or valleys, simulating springs, etc.

A few devices introduced in the literature, for instance, by Verplank and Georg (2011), have 1-DoF, measuring a linear position (or rotation) at the input and displaying force (or torque). They are known as 'haptic faders' or 'haptic knobs'. Examples of linear 1-DoF force-feedback faders are actuated sliders used in automated mixing consoles and in the *FireFaders* by Berdahl and Kontogeorgakopoulos (2013). Rotary 1-DoF devices include the *Haptic Knob* by Chu (2002), *the Plank* by Verplank et al. (2002), a low-cost haptic knob by Rahman et al. (2012), the *Haptic Capstans*, derived from the *FireFader* by Sheffield et al. (2016) and, more recently, the *TorqueTuner* by Kirkegaard et al. (2020) and Niyonsenga et al. (2022).

Among these 1-DoF force-feedback devices is the *TorqueTuner* by Kirkegaard et al. (2020) and Niyonsenga et al. (2022). It is singular, i.e., this module embeds haptics loop and effect presets in its microcontroller, and exposes input and output controls for mapping with external sound-synthesis engines, and comes in modular form factors as illustrated in Figure 2.

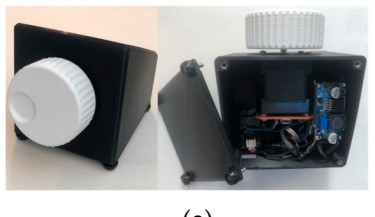 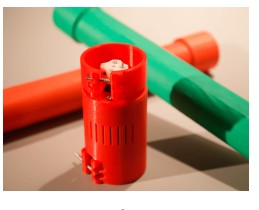 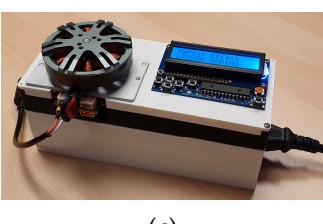

(**a**) (**b**) (**c**)

**Figure 2.** Modularity and evolution of the *TorqueTuner* from Kirkegaard et al. (2020) and Niyonsenga et al. (2022). The first two models, (**a**,**b**), are based on the Mechaduino platform. Model (**c**) is based on the Moteus platform due to the recent unavailability of the Mechaduino platform. (**a**) Standalone haptic knob (Mechaduino-based) by Kirkegaard et al. (2020). (**b**) T-stick adapter (Mechaduino-based) by Kirkegaard et al. (2020). (**c**) Workbench with presets (Moteus-based) by Niyonsenga et al. (2022).

### 2.1.2. How Many Further DoF Are Necessary for What?

There is no simple answer to this question, as devices with different numbers of DoF might be helpful in a given musical interaction. Therefore, the choice of a device should consider the intended use and the budget available for the project.

However, as previously shown, although simpler 1-DoF devices might be appropriate for specific interactions, e.g., plucking a string, many musical situations in the real world involve many DoF. Two examples include percussion and bowed-string instruments, specifically in the following scenarios:

- In percussion instrumental actions, the performer holds the stick at one end while the other is launched in a ballistic gesture toward the target. Rebound force is experienced by the player's hand, cf. Bouënard et al. (2010), Van Rooyen et al. (2017). Rebound force is transmitted along the stick to the hand, at which point it becomes a torque. Torques play an active role in percussion performance, for instance, influencing the timing of subsequent hits.

- In violin bowing, the performer holds the bow at the 'frog', while the hair–string interaction point varies away from the frog throughout a downward stroke. Several works in the literature, e.g., by Nichols (2000), O'Modhrain et al. (2000) and Tache et al. (2012), have tried to simulate bowing interactions, most of the time using three or fewer DoF. In Nichols's second version of VBow (Nichols 2002), 4-DoF were used. Indeed, as shown by Schoonderwaldt et al. (2007), forces such as bow weight pull the string along orthogonally to the bow, and both the application of pressure on the

string by the player and the rotation around the strings to select which string is bowed are exhibited as torques when translated from the bow–hair interaction point along the bow to the player's hand.

### 2.1.3. 3-DoF and 6-DoF Devices

Several commercial 3-DoF and 6-DoF devices exist. Although typically designed for industrial applications, many have been used in musical simulations.

Simulations of musical actions involving 6-DoF (force feedback in three translational and three rotational directions) are more complicated. Unlike 3-DoF devices, 6-DoF devices require more advanced mechanical technologies and complex computer modeling to integrate torque feedback seamlessly.

### 2.1.4. Multi-DoF Force-Feedback Devices

The Touch Back Keyboard by Gillespie (1992) with 1-DoF per key on eight keys and the MIKEY (Multi-Instrument active KEYboard) by Oboe (2006) with 1-DoF per key on three keys are two examples that illustrate the complexity of increasing the number of DoF in actuation to augment key-based instruments that already feature a large number of DoF in sensing.

One of the earliest examples of a force-feedback device designed to be used in sound and music interactions was the *coupleur gestuel retroactif*, developed by Florens (1978) at *ACROE*, in Grenoble, France. This was the first in a long series of devices explicitly designed for artistic/musical applications from the late 1970s to the 2010s, as reviewed by Cadoz et al. (2003) and Leonard et al. (2018). Although a few of these designs are mentioned here, it is hardly possible to overstate the contribution of *ACROE* to the area of force feedback and music, partially because these devices were conceived in the context of multi-pronged research on force feedback, alongside sound synthesis and animated images, as discussed by Cadoz et al. (1984). The iterations of force-feedback gesture transducers by Cadoz et al. (2003) go beyond the form factors of traditional musical instruments to enable multi-DoF digital musical instruments with customizable form factors and end effectors with up to 16-DoF. Their contributions in terms of novelty, quality and coherence over more than four decades are unique in computer music and haptics. Some of the most recent works from the group showed the feasibility of real-time, high-quality simulations of haptic/audio/visual environments controlled by force-feedback devices by Leonard et al. (2018), opening up the possibility for interactive multimedia performances that use force feedback.

### 2.2. Software Environments

When using force-feedback devices, one needs to define the behavior of the system comprising the device and the application context. For instance, when using a 3-DoF force-feedback controller, the feel of the device (forces output by the device) depends on the software model with which the device interacts. If the environment simulates a virtual wall, the force-feedback device end effector (e.g., a stylus) tends to stop when touching/trying to move through a wall (to a certain extent, depending on the characteristics of the simulation and the device used). If the environment consists of a pair of objects, one grounded to the floor and the other connected to the first one through a virtual spring, pushing the second one on the axis of the spring causes it to oscillate harmonically (if no friction is added to the environment). It is clear that the forces that the device outputs depend on both the device itself and the model being simulated.

Creating such models and virtual environments typically requires software tools to develop haptic simulations. Having been designed mainly for industrial or other non-artistic applications, such tools are not user-friendly for artists/musicians who do not possess strong programming expertise. Furthermore, they have limited capabilities when dealing with advanced sound generation/manipulation.

While many related works explore creative solutions for authoring haptic feedback, as reviewed by Schneider et al. (2017), Covaci et al. (2018) and Seifi et al. (2020), in this work, we focus on frameworks that couple force and sound feedback in musical applications.

### 2.2.1. Physical Modelling for Audio–Haptic Synthesis
#### *CORDIS-ANIMA*

Cadoz et al. (1993) pioneered the use of mass-interaction modeling for multisensory simulation. With *CORDIS-ANIMA*, designers designed physical behavior with scenes composed of interconnected masses, springs, non-linear links and friction elements. The resulting simulation is displayed through haptic, audio and visual outputs, all rendered with the same physical model. More recently, Villeneuve et al. (2015) introduced signal-modelling features.

#### *DIMPLE*

*DIMPLE* (*Dynamically Interactive Musically Physical Environment*) by Sinclair and Wanderley (2008) is a software framework that yields the creation of instrumental interactions using 3D objects with responsive behavior (visual, haptic and sound). In *DIMPLE*, a physical simulation of a virtual environment is constructed and can be manipulated by a force-feedback device. It uses *Open Sound Control* (*OSC*) by Wright and Freed (1997) and audio programming tools, such as *PureData* (*Pd*) by Puckette (1997), to create force-feedback-enabled virtual environments in *CHAI3D* by Conti et al. (2005). Objects in the environment can send back messages about their own properties or events, such as collisions between objects, using *Open Dynamics Engine* (*ODE*). These data can control events in sound synthesis or other media. *DIMPLE* has proven useful for multidisciplinary research in experimental psychology, multimedia, arts and computer music, e.g., in work by Erkut et al. (2008).

#### *Synth-A-Modeler*

*Synth-A-Modeler* (*SaM*) *Compiler* by Berdahl and Smith (2012) and *Designer* by Berdahl et al. (2016) together constitute an interactive development environment for designing force-feedback interactions with physical models. With *SaM*, designers interconnect objects from various paradigms (mass interaction, digital waveguides, modal resonators) in a visual programming canvas reminiscent of electronic schematics and mechanical diagrams and compile applications generated with the *Faust* digital-signal-processing (DSP) framework. *SaM Designer* does not support real-time visual rendering of models, and the possibilities of run-time modifications are limited to the tuning of object parameters.

#### *MIPhysics*

A more recent environment for prototyping force-feedback applications is *MIPhysics* by Leonard and Villeneuve (2020) (mi-creative.eu). Their collective *MI-Creative* uses mass-interaction physical modelling to create artistic applications generating physically-based sound synthesis, yielding fast prototyping of audio–haptic interactive applications; see Leonard and Villeneuve (2019). With *MIPhysics*, designers can script interactive simulations rendered with audio, haptic and visual feedback. Leonard and Villeneuve also developed a 1-DoF mass-interaction framework for *Faust* Leonard et al. (2019), aiming to design larger physical models but with no direct support for using haptic devices as input.

#### *ForceHost*

*ForceHost*, developed by Frisson et al. (2022), is a firmware generation toolkit for the *TorqueTuner*, as proposed by Kirkegaard et al. (2020). This toolkit expands the functionality of the *Faust* programming language toolkit to include modules for haptics, mappings, scriptable web-based user interfaces and sound synthesis, as illustrated in Figure 3.

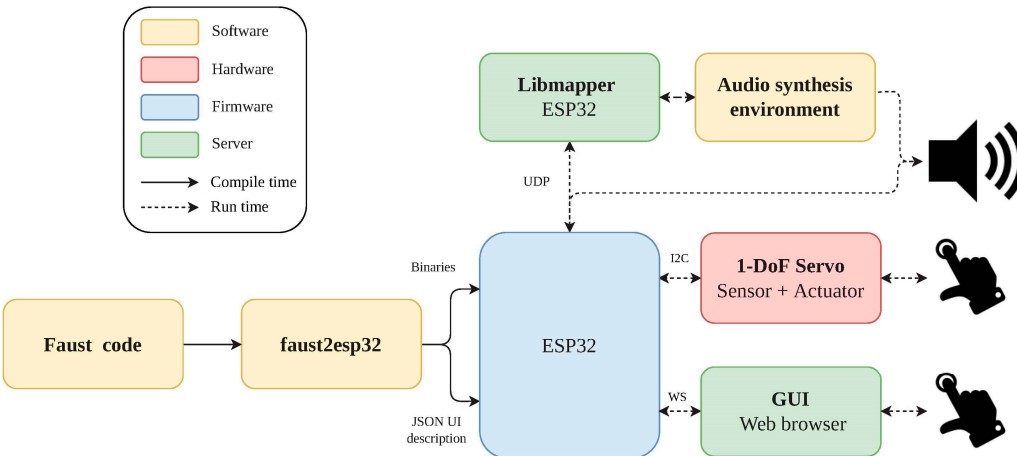

**Figure 3.** The architecture of *ForceHost*, from Frisson et al. (2022).

### 2.2.2. Force Feedback for Sample-Based Music Creation

Beamish et al. (2004) proposed *D'Groove* force-feedback-control techniques inherited from how disk jockeys (DJs) manipulate turntables. Frisson (2013, 2015) and colleagues investigated how force-feedback haptics support multimedia browsing, including for musical practices such as *comprovising* (or composing by improvising with) soundscapes by navigating through collections of sounds. They first devised prototypes to explore mappings between audio features and force-feedback controls with *DeviceCycle* by Frisson et al. (2010). They later created content-based force-feedback effects to browse collections of sounds, using motorized faders to recall sound effects applied to individual sounds in *MashtaCycle* by Frisson et al. (2013), adding friction when hovering over sound items with a haptic pointer and pulling the pointer towards the closest neighbor in a content-based similarity representation with *Tangible Needle* and *Digital Haystack* by Frisson et al. (2014).

## 3. Challenge

We identify four challenges in force-feedback musical instruments: modularity, replicability, affordability, and usability.

### 3.1. Modularity

Increasing the number of DoF for sensing and actuation introduces additional dimensions to the design space for interaction and display using force-feedback haptic devices. For instance, handheld manipulators of grounded force-feedback devices, such as the *3D Systems Touch* (formerly *SensAble Phantom Omni*), may feature 6-DoF for position sensing (3-DoF for translation, 3-DoF for orientation) and 3-DoF for actuation (motors actuating some joints of a serial arm, resulting in translations in 3D spaces), among other possible combinations of DoF, as illustrated by *Haptipedia*, which is an encyclopedia of force-feedback devices by Seifi et al. (2019). Larger numbers of DoF increase not only the potential complexity of interaction that the device can support but also the initial complexity of engineering the mechanical, electrical and computational architecture of these devices, for example the two force-feedback musical instruments in Section 2.1.4, the *Touch Back Keyboard* by Gillespie (1992) and the *MIKEY* (*Multi-Instrument active KEYboard*) by Oboe (2006). Rather than combining off-the-shelf devices with predefined form factors and enclosures, DMI designers may want to design their instruments by integrating their own selection of modules of DoF assembled in a mechanism that fits their instrument. *Force-feedback gesture transducers* by Cadoz et al. (2003) and *Probatio* by Calegario et al. (2020) are two use cases regarding challenges in modularity. The former went beyond the form factors of traditional musical instruments to enable multi-DoF force-feedback DMIs with customizable form factors and end effectors. However, it was designed by machine-engineered custom-ordered metal pieces, which are still hard to access for DMI designers, pre-dating contemporaneously

democratized 3D-printing solutions. The latter is a toolbox that enables DMI designers to combine various DoF and create different instruments adapted to various postures and metaphors of control that instrumentalists want to adopt while playing their instruments. Integration of force-feedback modules such as the *TorqueTuner* by Kirkegaard et al. (2020) into the *Probatio* toolbox is part of future work. It will pose challenges in supplying greater power for actuation and the distribution haptic parameters whilst maintaining haptic loops.

### 3.2. Replicability

Designers and players of DMIs face issues in being able to redesign and replay instruments that are not necessarily mass-produced and available off the shelf. DMIs may not have been designed for longevity, as studied by Morreale and McPherson (2017). The design and development process of DMIs may not have been documented in enough depth to be replicated, as reviewed by Calegario et al. (2021).

In addition to the issues mentioned above that are generic to DMIs, force-feedback haptic DMIs bear their own specific issues. Hardware connectors and ports eventually become obsolete (funds spent on devices). Software drivers are generally closed-source and clash with new APIs introduced along with OS generations. Operating systems manage real-time audio and haptic loops differently.

### 3.3. Affordability

The democratization of affordable open-hardware robotics platforms (e.g., *Arduino*, *ESP32*) and robotics application fields (electric devices for personal or light payload transportation, such as electric bikes and skateboards and drones) has enabled the prototyping of force-feedback haptics without the use of industrial facilities, such as in fabrication labs (fab labs). It has driven down the cost of components, particularly motors and electronic boards. In contrast, force-feedback devices are still not widespread, although this expanded availability of open-hardware components has reduced their costs. Over time, force-feedback device prices have gradually decreased from price ranges more typical of laboratory equipment and professional musical instruments (tens of thousands of dollars, including the *ERGOS TGR*, *MPB Technologies Freedom 6S*) to price ranges commensurate with computer peripherals and entry-level musical instruments (hundreds of dollars, including the *Novint Falcon*, *Haply Pantograph*, *TorqueTuner*). Such force-feedback devices are still not yet available in stores or homes in the manner that computer peripherals or entry-level musical instruments are. Leonard et al. (2020) argued that nowadays, affordable force-feedback devices are sufficient for "*thinking* and *designing* dynamic coupling with virtual musical instruments, but they do not yet entirely allow qualitative *feeling* of this coupling".

### 3.4. Usability

Seifi et al. (2020) reviewed the challenges met by novice force-feedback haptic designers ("hapticians") to create applications with 1-DoF devices throughout the 'Student Innovation Challenge' at the World Haptics Conference in 2017. The authors concluded that novice hapticians have several needs for haptic design: theoretical and practical guidelines, tools for infrastructure and content, and an ecosystem of authoring tools. In addition, expert hapticians have been adopting design practices and tools from related fields to generate content through audio and visual modalities.

Novice and expert designers of non-audio–haptic applications face similar challenges when designing DMIs that integrate force feedback and sound synthesis. These challenges go beyond the lack of design guidelines and tools because they also require support for both audio and haptic modalities. Authoring tools for designing for both audio and haptic modalities are scarce. To our knowledge, only *GENESIS* by Villeneuve et al. (2015), *Synth-A-Modeler Designer* by Berdahl et al. (2016), and *ForceHost* by Frisson et al. (2022) exist, which propose physical modelling metaphors or signal-based approaches. When authoring tools support only one modality, either audio or haptic, designers must devise strategies to synchronize streams, which often requires ad hoc development.

## 4. Opportunities

We identify three opportunities for further research in FF&M:

- embedding audio and haptic software into hardware modules
- networking multiple modules with distributed control
- authoring with audio-inspired and audio-coupled tools.

### 4.1. Embedding

To address challenges related to replicability and usability, we propose the integration of audio and haptic processing and authoring in microcontrollers. This involves embedding haptic loops, as demonstrated in the *TorqueTuner* by Kirkegaard et al. (2020), as well as embedded drivers and web-based control panels, as illustrated in *ForceHost* by Frisson et al. (2022). By embedding these software components directly into microcontrollers that interface with hardware components, audio–haptic DMIs no longer depend on third-party operating systems to maintain and synchronize audio and haptic loops. This also makes them less sensitive to APIs and peripheral connectors changes because the drivers and control panels are on-board and can communicate with third-party computers using interoperability protocols, such as *OSC*. Alternatively, they may only require a default web browser for authoring.

### 4.2. Networking

To overcome challenges in modularity and replicability, we propose to network audio and haptic modules. Beyond reusing off-the-shelf force-feedback devices, audio–haptic DMI designers now have the opportunity to combine force-feedback modules, such as the *Firefader* by Berdahl and Kontogeorgakopoulos (2013) (one translational DoF) and the *TorqueTuner* by Kirkegaard et al. (2020) (one rotational DoF), and instead devise their own modular user interface, as with *Probatio* by Calegario et al. (2020). Further research is needed to understand how to effectively arrange all audio–haptic streams and ensure their level of synchronicity. A promising avenue for networks of embedded modules is to explore the mapping of signals using solutions such as *libmapper*, as proposed by Malloch et al. (2013), along with its web-based authoring tool *webmapper*, as demonstrated by Wang et al. (2019). Specifically, sparse event-based control signals can be used for mapping rather than embedding audio or haptic loops in each module.

### 4.3. Authoring

To overcome modularity, replicability and usability challenges, we propose further developing audio-inspired and audio-coupled force-feedback haptic-authoring tools.

Audio-inspired haptic-authoring tools should reuse well-established features from audio-authoring tools, such as digital audio workstations where graphical representations of waveforms and transfer functions are commonplace, where interoperability protocols, such as the *Musical Instrument Digital Interface* (*MIDI*), *MIDI Polyphonic Expression* (*MPE*) and *OSC* are well established, and where APIs for audio effects and synthesizers plugins yield enrichment of the audio design space. Further research is needed to define what would be suitable interoperability protocols for force-feedback haptics, similar to how *TUIO* by Kaltenbrunner et al. (2005) and Kaltenbrunner and Echtler (2018) expanded *OSC* for tangible user interfaces and what plugin-API would be suitable for force-feedback haptics that could also be implemented with the networking of embedded modules.

Audio-coupled haptic-authoring tools should facilitate the design of audio and haptic feedback with a unified system, sharing one scripting language or one visual programming metaphor for the designs for both modalities. For instance, *ForceHost* by Frisson et al. (2022) explored how the *Faust* DSP programming language could be employed to unify the description of audio and haptic applications, including their control through auto-generated web-based user interfaces (see Section 2.2.1).

### 5. Conclusions

This paper reviewed the literature on research works that combine music and force-feedback haptics. We discussed the limitations of these works and elicited the main challenges in current applications of force feedback and music: modularity, replicability, affordability and usability.

We identified opportunities in future research into force feedback and music, namely embedding audio and haptic software into hardware modules, networking multiple modules with distributed control and authoring with audio-inspired and audio-coupled tools. Our review has been accompanied by examples of our recent work, which includes the development of an affordable, open-source and self-contained 1-DoF rotary force-feedback device for musical applications, known as the *TorqueTuner*, as described by Kirkegaard et al. (2020). Furthermore, we have also embedded audio and haptic processing/authoring in module firmware through *ForceHost*, as presented by Frisson et al. (2022).

**Author Contributions:** Conceptualization, methodology, investigation, resources, writing, funding acquisition: C.F. and M.M.W. All authors have read and agreed to the published version of the manuscript.

**Funding:** This research was funded by Natural Sciences and Engineering Research Council grant number RGPIN-2019-04551; Ministère de l'Économie et de l'Innovation du Québec grants number 19-22-PSOv2a-45608 and 19-22-PSOv2a-45604. The APC was funded by M.M.W. who is Special Issue Editor for Special Issue Feeling the Future—Haptic Audio (https://www.mdpi.com/journal/arts/special_issues/feeling_the_future (accessed on 8 May 2023)) where this article is published.

**Acknowledgments:** The authors would like to thank past collaborators for our discussions and collaboration over the years, most notably Albert-Ngabo Niyonsenga, Jean-Loup Florens, Marcello Giordano, Mathias Bredholt, Mathias Kirkegaard, Olivier Tache, Stephen Sinclair. Part of this work has been supported by multiple grants from the Natural Sciences and Engineering Research Council of Canada (NSERC), through its Discovery, RTI (Research Tools and Instruments) and New Opportunities programs.

**Conflicts of Interest:** The authors declare no conflict of interest.

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
