# Peer review of "Challenges and Opportunities of Force Feedback in Music"

_arts, 2010_

Round 1
Reviewer 1 Report
The field of haptic, non-vibratory, force feedback for musical applications has been the subject of scientific investigations and technological developments over the last decades. In this paper, that reads like a paper half-way between a review paper and an opinion paper, the authors give an insightful review of past and present research in this topic, pointing their limits, identifying challenges and proposing solutions to overcome these challenges.
I have but minor comments that I list below.
- acronyms and abbreviations must be defined (most are not, some are defined but not at first use): "FF&M" (L57), "DoF" (L62), "ACROE" (L78), "FF" (173), "OSC" (L201 -> this one also requires a more complete definition than just spelling the acronym, see also L208 "OpenSoundControl" that is not defined)
- please define "slice motors" (L81)
- Need for precisions at some places: are they translation or rotation (and if applicable: which axes are involved?) DoFs for the Novint Falcon (L63)?
- Need for providing orders of magnitude of the volume of workspace (L70): how large are the zones allowed by each of the devices the authors describe?
- Need for providing orders of magnitude of the output forces (L71): how lower are the "lower output forces"?
- Brands and models are first italicized (L61-68), then no more (L69-75), then both (L77-78), etc.: I'd recommend consistency throughout the manuscript. (I'd personally go for italics everywhere)
- Figure 1: it is not clear where each device described in the caption can be found in the image. For clarity, please circle each device with a specific color, it will be much easier to identify each device for the reader who are not familiar with them.
- sentence "spanning from the later 70's to the 2010's" reads strange, since the reference for this is dated 2003. Is there maybe a more recent reference?
- please consider replacing "simulating" with "simulates" (L175)
- I'm not sure I understand "wither" on L88, is it "either"?
- Reference "Leonard & Villeneuve 2019" on L461-462 is missing a book title (+ presumably editors' names)
Author Response
We thank Reviewer 1 (R1) for their insightful suggestions that we address below.
R1: Brands and models are first italicized (L61-68), then no more (L69-75), then both (L77-78), etc.: I'd recommend consistency throughout the manuscript. (I'd personally go for italics everywhere)
We harmonized all brands and models and tools (software and hardware) with italics (\emph{}), except in biographies, for these terms, in alphabetical order:
- 3D Systems
- ACROE
- Arduino
- CHAI3D
- coupleur gestuel retroactif
- DeviceCycle
- DIMPLE
- ERGOS
- ESP32
- Falcon
- FAUST
- FireFader
- ForceHost
- Force Feedback Gesture Transducers
- Freedom 6S
- GENESIS
- Haply
- Haptic Capstans
- haptic faders
- Haptic Knob
- Haptipedia
- Jitter
- GEM
- Logitech
- libmapper
- MashtaCycle
- Max
- MI-Creative
- MIDI Polyphonic Expression (MPE)
- MIKEY (Multi-Instrument active KEYboard)
- MIPhysics
- MPB Technologies
- Musical Instrument Digital Interface (MIDI)
- Novint
- Omni
- OpenHaptics
- Open Dynamics Engine (ODE)
- Open Sound Control (OSC)
- Phantom Desktop and Omni and Premium
- Pantograph
- Pd
- Probatio
- PureData
- the Plank
- SensAble
- Synth-A-Modeler (SaM) Compiler and Designer
- Tangible Needle and Digital Haystack
- TorqueTuner
- Touch Back Keyboard
- Touch X
- TUIO (has no official debreviation, closest is: Tangible User Interface Input Output)
- webmapper
- WingMan
R1: acronyms and abbreviations must be defined (most are not, some are defined but not at first use): "FF&M" (L57), "DoF" (L62), "ACROE" (L78), "FF" (173), "OSC" (L201 -> this one also requires a more complete definition than just spelling the acronym, see also L208 "OpenSoundControl" that is not defined)
We defined acronyms and abbreviations:
- force-feedback and music (FF\&M) in abstract L5
- Degree-of-Freedom (DoF) in abstract L9 (already before review)
- the Association pour la Création et la Recherche sur les Outils d'Expression (ACROE) L101
- force feedback (FF) in abstract L2
- Open Sound Control (OSC), a network protocol created by \cite{wright_open_1997} initially for communicating with sound synthesizers L228
- Musical Instrument Digital Interface (MIDI) L381
- MIDI Polyphonic Expression (MPE) L381
R1: please define "slice motors" (L81)
We corrected "slice motors" into "sliced motors" based on a re-read of \cite{cadoz90b} and defined it as follows based on \cite{cadoz90b}:
- \emph{sliced motors} (motors sharing a single magnetic polarization circuit for use as motor modules in a FF keyboard) L105
R1: Need for precisions at some places: are they translation or rotation (and if applicable: which axes are involved?) DoFs for the Novint Falcon (L63)?
We added more details (**...**) to the following sentence:
Some of these devices measure positions in 3 DoF (e.g. Novint Falcon **with 3 DoFs in translation**), while others measure position in 6 DoF **(for instance 3 DoFs in translation and 3 DoFs in rotation between the stylus and the arm of the end effector)**, whilst still providing a 3 DoF force output, e.g. \emph{3D Systems's Touch X}, formerly the \emph{SensAble's Phantom Desktop}.
R1:
- Need for providing orders of magnitude of the volume of workspace (L70): how large are the zones allowed by each of the devices the authors describe?
- Need for providing orders of magnitude of the output forces (L71): how lower are the "lower output forces"?
We think that this level of detail might be too high our paper, however here is how we can address this recommendation at L85:
The Touch X, Phantom Premium and Freedom 6S are three examples of variable workspace and force distributions, based on data collected for and categorized in Haptipedia by \cite{seifi_haptipedia_2019}: %
%
the Touch X has a small translational workspace of $16*12*12$ ${\textrm{cm}}^3$ and large rotational workspace of $360*360*180$ ${\deg}^3$ and outputs translational forces of 7.9 N (peak) and 1.75 N (constant),
% https://haptipedia.org/?device=SensAble3DOF1999d
% Motion range (Workspace size)
% Translational workspace size Small
% Translational workspace size - detail D1 (16 cm), D2 (12 cm), D3 (12 cm)
% Rotational workspace size Large
% Rotational workspace size - detail D1 (360 °), D2 (360 °), D3 (180 °)
% Translational force (N) - peak 7.9 N
% Translational force (N) - constant 1.75 N
%
the Phantom Premium has a large translational workspace of $82*59*42$ ${\textrm{cm}}^3$ and large rotational workspace of $330*330*220$ ${\deg}^3$ and outputs translational forces of 22 N (peak) and 3.00 N (constant),
% https://haptipedia.org/?device=SensAble6DOF1999
% Motion range (Workspace size)
% Translational workspace size Large
% Translational workspace size - detail D1 (82 cm), D2 (59 cm), D3 (42 cm)
% Rotational workspace size Large
% Rotational workspace size - detail D1 (330 °), D2 (330 °), D3 (220 °)
% Translational force (N) - peak 22 N
% Translational force (N) - constant 3.00 N
%
the Freedom 6S has a medium translational workspace of $33*22*17$ ${\textrm{cm}}^3$ and medium rotational workspace of $340*170*130$ ${\deg}^3$ and outputs translational forces of 2.5 N (peak) and 0.60 N (constant).
% https://haptipedia.org/?device=MPB6DOF2000
% Motion range (Workspace size)
% Translational workspace size Medium
% Translational workspace size - detail D1 (33 cm), D2 (22 cm), D3 (17 cm)
% Rotational workspace size Medium
% Rotational workspace size - detail D1 (340 °), D2 (170 °), D3 (130 °)
% Translational force (N) - peak 2.5 N
% Translational force (N) - constant 0.60 N
R1: Figure 1: it is not clear where each device described in the caption can be found in the image. For clarity, please circle each device with a specific color, it will be much easier to identify each device for the reader who are not familiar with them.
We replaced the figure with a more up to date version with a greater variety of devices.
We used https://ff.cx/latex-overlay-generator/#/v0.0.1 to generate accessible callouts around each device referenced with a letter and cited in the caption.
We used letters instead of colors to identify devices, as it is hard to find a colorblind-safe color map for categorical data with 12 classes (14 items).
R1: sentence "spanning from the later 70's to the 2010's" reads strange, since the reference for this is dated 2003. Is there maybe a more recent reference?
We looked for a newer reference in this collection:
https://hal.science/search/index/?q=structName_s%3A%22ACROE+-+Ing%C3%A9nierie+de+la+Cr%C3%A9ation+Artistique%22&rows=30&sort=producedDate_tdate+desc
The newest in this list relevant to FF&M is from 2018 and part of our references (Leonard2018).
We updated the sentence to contain:
"as reviewed by \cite{cadoz_acroe_2003,Leonard2018}"
R1: please consider replacing "simulating" with "simulates" (L175)
Indeed, the sentence now contains:
"If the environment simulates a virtual wall"
R1: I'm not sure I understand "wither" on L88, is it "either"?
We have replaced "wither" by "either".
R1: Reference "Leonard & Villeneuve 2019" on L461-462 is missing a book title (+ presumably editors' names)
We have fixed the related BibTeX entry to reveal that this work was presented at the International Workshop on Haptic Audio Interaction Design (HAID2019) with proceedings.
ADDITIONAL UPDATES
We have added the following sentence at the end of Challenge: Affordability:
\cite{leonard_multisensory_2020a} argue that affordable force-feedback devices are nowadays sufficient for ``\textit{thinking} and \textit{designing} dynamic coupling with virtual musical instruments, but they do not yet entirely allow qualitative \textit{feeling} of this coupling''.
Reviewer 2 Report
The manuscript presents a survey of force feedback in musical applications. The survey covers the basics of force feedback, prior uses of force feedback in musical applications, as well as challenges and opportunities identified by the authors.
The manuscript is well written, clearly structured and easy to follow. It has a relatively narrow scope, given that force feedback has received limited attention in musical applications as stated by the authors, but the works covered appears to be exhaustive and largely complete. The survey also includes a higher-level analysis with relevant discussions of challenges and opportunities. Overall I am satisfied with the survey and I believe that it is nearly ready for publication. I have a few suggestions to that could improve it:
- The abstract and introduction are very clear about the limited scope of the survey, which excludes a larger body of work on vibrotactile feedback in musical application. While it is acceptable (and necessary) to define this scope, I wonder if it would be worthwhile to have at least a brief section that survey prior work using vibrotactile feedback, and perhaps limitations of such work that would justify the use of force feedback. I also wonder if something could be said about the importance of force feedback in traditional musical instruments.
- Similarly, I was surprised to see mentions of “voice coils” and a “woofer” in Section 2.1.2. It would perhaps be best to avoid such mentions if the scope is limited to force feedback devices.
- The abstract and conclusion mention that the authors’ prior is used to illustrate the review. This didn’t come across as clearly as it could. The discussion of opportunities, for example, seems to use the authors’ prior work as well as many others for example. I would suggest either making this use of the authors’ prior work more explicit, or simply discussing it without mentioning that it is the authors’ prior work.
- I would suggest looking at the work of Karon MacLean for other examples of force feedback in media control and musical instruments. DOI 10.1145/985692.985734, in particular, seems relevant.
Author Response
We thank Reviewer 2 (R2) for their insightful suggestions that we address below.
R2: I wonder if it would be worthwhile to have at least a brief section that survey prior work using vibrotactile feedback, and perhaps limitations of such work that would justify the use of force feedback. I also wonder if something could be said about the importance of force feedback in traditional musical instruments.
- We added these sentences right at the beginning of the Introduction:
Digital Musical Instruments (DMI) feature high-resolution gesture sensing and audio output, but poor gestural and bodily feedback, compared to traditional acoustic instruments, which passively produce and transfer: vibration from strings to fingers;kinesthetic feedback from drums to fingers, hands, forearms and arms of drummers; coupling of air columns between wind instruments and their players. In this article, we examine how DMIs can produce dynamic feedback to the sense of touch of their players, that is through haptic feedback.
- We rephrased and complemented the following sentence at L18:
Touch or vibrotactile feedback is discussed in length in other papers part of the same special issue ``Feeling the Future—Haptic Audio'' as our article, as well as in a recent reference edited by \cite{PapettiSaitis2018}.
- We added the following sentence a the end of the now second paragraph of the Introduction that focused on FF vs vibrotactile:
While vibrotactile feedback mainly stimulates the skin, force-feedback extends the possibilities of stimulation to a larger set of the body, with the capability of responding finely to subtle movements from our limbs. And musicians employ as much of their whole body as they are able to while playing music.
R2: Similarly, I was surprised to see mentions of “voice coils” and a “woofer” in Section 2.1.2. It would perhaps be best to avoid such mentions if the scope is limited to force feedback devices.
We removed the following sentence:
Sometimes, mechanically simpler solutions might be effective as well. For instance, the \emph{haptic drum}, presented by \cite{Berdahl2008a}, uses a woofer (1-DoF) to help create drum rolls.
We kept the following sentence because we believe that this work illustrates well many-DOFs applications with voice coil actuators producing displacements of several centimeters:
Simulation of such actions can be achieved with voice coils, as impressively done by \cite{Rooyen17}.
R2: The abstract and conclusion mention that the authors’ prior is used to illustrate the review. This didn’t come across as clearly as it could. The discussion of opportunities, for example, seems to use the authors’ prior work as well as many others for example. I would suggest either making this use of the authors’ prior work more explicit, or simply discussing it without mentioning that it is the authors’ prior work.
We chose to make our joint prior work more explicit:
- we removed the figure showing DIMPLE, another prior work that involves only one of the authors of this article
- we added figures for TorqueTuner and ForceHost.
R2: I would suggest looking at the work of Karon MacLean for other examples of force feedback in media control and musical instruments. DOI 10.1145/985692.985734, in particular, seems relevant.
We have added the following sentence at the beginning of subsubsection Force-feedback for sample-based music creation:
\cite{beamish_manipulating_2004} have proposed with \emph{D'Groove} force-feedback control techniques inherited from how disk jockeys (DJs) manipulate turntables.
ADDITIONAL UPDATES
We have added the following sentence at the end of Challenge: Affordability:
\cite{leonard_multisensory_2020a} argue that affordable force-feedback devices are nowadays sufficient for ``\textit{thinking} and \textit{designing} dynamic coupling with virtual musical instruments, but they do not yet entirely allow qualitative \textit{feeling} of this coupling''.